# Efficient Reject Options for Particle Filter Object Tracking in Medical Applications

**DOI:** 10.3390/s21062114

**Published:** 2021-03-17

**Authors:** Johannes Kummert, Alexander Schulz, Tim Redick, Nassim Ayoub, Ali Modabber, Dirk Abel, Barbara Hammer

**Affiliations:** 1Machine Learning Group, Bielefeld University, 33619 Bielefeld, Germany; aschulz@techfak.uni-bielefeld.de (A.S.); bhammer@techfak.uni-bielefeld.de (B.H.); 2Institute of Automatic Control, RWTH Aachen University, 52074 Aachen, Germany; T.Redick@irt.rwth-aachen.de (T.R.); D.Abel@irt.rwth-aachen.de (D.A.); 3Department of Oral and Maxillofacial Surgery, University Hospital RWTH Aachen, 52074 Aachen, Germany; nayoub@ukaachen.de (N.A.); amodabber@ukaachen.de (A.M.)

**Keywords:** secure object tracking, reject option, particle filtering, assisted surgery

## Abstract

Reliable object tracking that is based on video data constitutes an important challenge in diverse areas, including, among others, assisted surgery. Particle filtering offers a state-of-the-art technology for this challenge. Becaise a particle filter is based on a probabilistic model, it provides explicit likelihood values; in theory, the question of whether an object is reliably tracked can be addressed based on these values, provided that the estimates are correct. In this contribution, we investigate the question of whether these likelihood values are suitable for deciding whether the tracked object has been lost. An immediate strategy uses a simple threshold value to reject settings with a likelihood that is too small. We show in an application from the medical domain—object tracking in assisted surgery in the domain of Robotic Osteotomies—that this simple threshold strategy does not provide a reliable reject option for object tracking, in particular if different settings are considered. However, it is possible to develop reliable and flexible machine learning models that predict a reject based on diverse quantities that are computed by the particle filter. Modeling the task in the form of a regression enables a flexible handling of different demands on the tracking accuracy; modeling the challenge as an ensemble of classification tasks yet surpasses the results, while offering the same flexibility.

## 1. Introduction

Novel technological developments have led to great advances in computer assisted surgery, whereby image-guided support of surgeons plays a particularly important role besides direct robotic assistance [1]. Some advances can be observed in recent years, including, among other approaches, dedicated open source platforms to advance and evaluate technical progress [2]. Assistive systems often rely on visual information, such as videos or depth images, and imaging sensors play a crucial role for a wide array of medical domains. One such domain with a potentially high benefit constitutes the field of osteotomy, where the exact position and pose of the bone has to be known precisely. In this context, current practice is to use markers, such as 3D-printed templates, to assist the surgeon [3]. However, the design and production of these templates constitutes a significant overhead that could be avoided by markerless tracking systems. Such intelligent tracking methods enable the reliable identification or segmentation of the images into semantically meaningful parts, e.g., relevant organs, bone structure, or malignant parts, and tracking these over time.

Several approaches address optical tracking in the medical domain, such as models that are based on optical flow [4], or probabilistic frameworks [5]. In the last decade, particle filtering became one of the most prominent methods for reliable positioning and tracking in the medical domain and beyond [6,7]. Particle filtering basically constitutes a technique for an efficient Monte Carlo simulation to solve filtering problems that arise when internal states of dynamic systems, such as the location of a specific constituent in an image, need to be estimated based on partial observations that are subject to sensor noise. Thereby, a crucial part of the model is the design of a suitable probability model that captures the true underlying probability but is sufficiently easy to compute. There exists extensive work regarding appropriate probabilistic models in continuous and discrete settings [8,9] as well as extensive research on the accuracy of the result and how to arrive at unbiased estimators [10,11,12,13]. Usually, real-time performance is required, such that computationally complex models are prohibited and considerable work addresses computation schemes on the edge [14].

Popular models can typically incorporate some form of generic measurement noise, yet major disturbances, such as occlusions, pose a challenge. Hence, it is a matter of ongoing research as to how to design robust tracking schemes that enable a re-detection of the object after it has been lost and realize a larger robustness and invariance to noise, as an example, by means of multiple correlation filters or robust representations, such as hidden layers of deep networks [15,16,17]. However, in realistic scenarios, it is unavoidable that tracked objects are lost in some settings, as an example, in the case of a total occlusion of the tracked object. In critical settings, such as assisted surgery, it is desirable to equip tracking algorithms with the notion of a reject option—meaning that tracking algorithms are capable of detecting on their own if the tracking is no longer reliable and a major discrepancy of the tracked object and the result of the tracking algorithm is to be expected. In such cases, an autonomous or assisted re-initialization of the tracking algorithm could be done and major harm due to a lost tracking be prevented. Work in this direction exists that focuses on rejecting outlier measurements [18], which uses non-parametric kernel density estimation. The latter is well known to be impractical already for a medium number of dimensions [19]. In contrast, the sum of likelihoods of updated particles is used for a decision on rejecting in [20]. The authors of [21] mitigate the effect of corrupted observations by updating only a subset of the particles, where an observation is considered to be corrupted if the largest particle likelihood is below a pre-defined threshold. We consider this detection mechanism, which is also investigated by Chow [22] in a more general setting, as a baseline in our evaluation. In this contribution, we investigate the challenge of how to equip a recent state-of-the-art particle filtering technology for object tracking, which is based on numerically stable representations of particle weights in the logarithmic domain [23], by efficient possibilities to equip the tracking result with a notion of reliability in the form of a reject option, which filters too large deviations of the tracked quantities from their true values. We investigate this, in particular, with respect to changing configuration settings of the particle filter.

The question of how to enhance models with a notion of their reliability and an according reject constitutes an open research challenge for more general models than particle filters, as considered in our approach. For example, in classification tasks, the notion of selective classification or classification with a reject option has already been introduced quite early on [22]. It has been shown that a simple threshold strategy yields an optimum reject in the sense of cost optimization, provided that the true underlying probability is known. The latter is usually not the case, and the models inferred from data provide approximations of the true values only, provided probabilistic, rather than frequentist, modeling takes place. As a consequence, it has been investigated, in how far deterministic models can be extended to incorporate a reject option by suitable loss modeling [24], how confidence values can be approximated by suitable alternatives that are provided by the given model [25], and how reject options can be learned in parallel to the task at hand [26]. Note that statistical frameworks exist—such as conformal prediction [27]—which compute confidence values from a given model and data in such ways that strong mathematical guarantees hold. Yet, these approaches are usually time consuming and they cannot be used in an online tracking task where real-time conditions hold.

In this contribution, we address the question of how to equip a state-of-the art particle filter for tracking by efficient schemes that implement a reliable reject option to detect cases where the tracking is invalid. Because particle filters are based on a probabilistic modeling and, hence, provide likelihood values, a natural baseline builds a reject option on a global threshold strategy: settings are rejected if the likelihood is too small, following the strategy, as investigated by Chow [22]. It turns out that the results of this threshold strategy are not satisfactory for a benchmark problem from the domain of assisted surgery. We propose modelling the problem as a machine learning task, and we propose two possible ways to formalize the problem: as a regression task that predicts the amount of displacement, or as an ensemble of classification tasks that detect settings for which the misplacement is larger than a given threshold. We will formalize these two approaches and evaluate those in a realistic benchmark and different settings of the particle filter.

## 2. Materials and Methods

### 2.1. Experimental Prerequisites

A novel approach to jaw reconstruction surgery utilizes bone cut from the patient’s pelvis (Figure 1a) to create a graft for the jaw (Figure 1b) with low risk of getting rejected by the body [28]. For the cutting procedure a custom 3D model is created as a template for the surgeon. So far, this model was printed and then placed on the bone, which is expensive and time consuming. In a new approach [29], a robotic arm is equipped with a depth camera to track a representation of the template in the depth image while using a particle filter [23], and a projector to display cutting outlines onto the bone (Figure 2) during surgery. However, this new approach leads to the risk that the projection is off when the tracking is inaccurate.

The projection cannot be static, since displacements of the body are unavoidable during surgery. Hence, a tracking of the shape and an according adjustment of the projection is necessary. There exist several reasons why tracking can be wrong—as an example, a major challenge consists in the fact that the tracking target might be partially or fully occluded by the surrounding tissue or a person.

Our goal is to find a method that identifies when the track is inaccurate, so the surgeon can be notified to stop cutting, and a manual reset may be conducted, if needed. Thereby, we want to find a technology that works for different parameterizations of the particle filter that we will introduce in Section 2.2, such that an easy transfer in between different settings becomes possible. Because it might be specific to the situation, which deviation of the tracking from the ground truth is crucial, we also consider different degrees of inaccuracy that will be explained in Section 2.4. This way, our method can also be applied to other setups that utilize particle filter tracking.

For this purpose, we utilize the data and particle filter implementation from the study [29]. In the following, we first summarize the according particle filter methodology before we then provide more details on the used data sequences.

### 2.2. Particle Filter Tracking

Assume a depth image at time *t* is denoted
(1)$$Im_{t}=\left(\begin{array}{ccc} x_{1,1} & \cdots & x_{1,n}\\ \vdots & \ddots & \vdots \\ x_{m,1} & \cdots & x_{m,n} \end{array}\right)$$
where *m* and *n* are the width and height of the depth image and $x_{i,j}$ the distance to the camera of the pixel at position i,j. The basic objective of the particle filter tracking is to find the pose
(2)$${\hat{\rho }}_{t}=\left(\begin{array}{c} {\hat{p}}_{t}\\ {\hat{q}}_{t}\\ {\hat{\nu }}_{t}\\ {\hat{\nu }}_{t}^{rot} \end{array}\right)\in R^{14}$$
consisting of position ${\hat{p}}_{t}$ as 3D coordinates in the room, and orientation ${\hat{q}}_{t}$ as a quaternion, for the 3D model of the graft that has the biggest pixel overlap to the depth image $Im_{t}$ at time *t*. ${\hat{\nu }}_{t}^{trans}$ is the current translational velocity of the particle and ${\hat{\nu }}_{t}^{rot}$ the current rotational velocity. The tracking algorithm that has been implemented and evaluated in the approach [29] consists of the following steps: it is manually initialized at approximately the correct position. Subsequently, within an iterative loop, we predict *K* possible new poses $P_{t+1}=\left(\rho_{t+1}^{1},\cdots ,\rho_{t+1}^{K}\right)$ (particles) by sampling according to a noise distribution
(3)$$\rho_{t+1}∼n\left(\rho_{t}\right)=\left(\begin{array}{c} p_{t}+\nu_{t+1}^{trans}dt\\ q_{t}+0.5\left(q_{t}\nu_{t+1}^{rot}\right)dt\\ vf\cdot \nu_{t}^{trans}+N\left(0,n^{trans}\right)dt\\ vf\cdot \nu_{t}^{rot}+N\left(0,n^{rot}\right)dt \end{array}\right)$$
with the parameters for the translational noise $n^{trans}$, rotational noise $n^{rot}$, and a velocity factor vf. N refers to the normal distribution. The new velocities are obtained by adding gaussian noise to the old ones. Additionally, the velocity is assumed to be decaying due to a loss of energy that is modeled by the multiplication of the velocity factor. Using the new velocities, the new position and orientation are calculated by applying the backward Euler integration method. Note that the new orientation is the *Nlerp* between our old orientation and the new orientational velocity. For each new particle, we generate an expected depth image $Im_{\rho_{t+1}^{k}}$ that is based on simple geometric principles, and we calculate the weight
(4)$$w_{\rho_{t+1}^{k}}\left(Im_{t+1},Im_{\rho_{t+1}^{k}}\right)=\sum_{i=1,j=1}^{m,n}ll\left(Im_{t+1}\left(i,j\right),\;Im_{\rho_{t+1}^{k}}\left(i,j\right)\right),$$
where ll is the log likelihood estimating how likely the observed pixel $x_{o}$ fits the expected pixel $x_{e}$ and it is calculated as
(5)$$ll\left(x_{o},x_{e}\right)=\left\{\begin{array}{cc} 0 & \mathrm{if}\;x_{e}\;\mathrm{invalid}\\ ln(w^{uni}*(1/\left(\hat{d}-\overset{ˇ}{d}\right)+ & \\ w^{gaus}*N\left(x_{e},d_{f}\cdot x_{e}^{2}+bn\right)+ & \\ w^{exp}*\left(\lambda e^{-\lambda x_{o}}\right)/\left(1-e^{-\lambda x_{e}}\right)) & \mathrm{if}\;x_{e}\geq x_{o}\\ ln(w^{uni}*(1/\left(\hat{d}-\overset{ˇ}{d}\right)+ & \\ w^{gaus}*N\left(x_{e},d_{f}\cdot x_{e}^{2}+bn\right)) & \mathrm{if}\;x_{e}<x_{o}\\ ln(w^{uni}*\left(1/\left(\hat{d}-\overset{ˇ}{d}\right)\right) & \mathrm{if}\;x_{o}\;\mathrm{invalid} \end{array}\right.$$
where $w^{uni},\;w^{gaus},\;w^{exp}$ are configurable as uniform, Gaussian, and exponential weight, and the camera intrinsic values $\hat{d},\;\overset{ˇ}{d}$ as maximum and minimum depth value, $d_{f}$ as depth factor, and bn as the camera’s base noise. In the first case, the expected depth at this pixel is invalid, i.e., it belongs to the background, therefore any observation fits. In the second case, occlusion is a possible reason why the observation is closer than the expectation. This possibility is modeled with an additional exponential probability term as compared to the third case. In the case of an invalid observation, the uniform weight is taken as a random measurement (case 4). The tracking pose ${\hat{\rho }}_{t+1}$ is then determined by the particle with the maximum log likelihood
(6)$${\hat{\rho }}_{t+1}=\rho_{t+1}^{{\mathrm{argmax}}_{k}w_{\rho_{t+1}^{k}}}.$$

See the reference [23] for details.

### 2.3. Tracking Scenario

As a concrete tracking scenario that is representative for assisted surgery, we use a *rosbag* [30] that was recorded during a test surgery on a cadaver at University Hospital RWTH Aachen. This bag contains the depth image data during the whole surgery as a *sensor_msgs/PointCloud2* [31] and a tracking output as *geometry_msgs/Pose* [32] (Figure 3). Because this output was under human supervision and manually reset and corrected, we can utilize it as our ground truth ${\tilde{\rho }}_{t}$. This allows for us to rerun the tracking while replaying the bag to record different outputs of the tracking algorithm in different configurations.

For each run, we save the parameter configuration $X_{\mathrm{conf}}\in R^{7}$ which contains: translational noise ($n^{trans}$), rotational noise ($n^{rot}$), velocity factor (vf), exponential weight ($w^{exp}$), gaussian weight ($w^{gaus}$), uniform weight ($w^{uni}$), and particle count (*K*). Table 1 shows the chosen value ranges.

Additionally, we record the particle filter output $X_{\mathrm{out}}\in R^{4}$ that contains the maximum log likelihood $\hat{ll}=\max_{k}w_{\rho_{t}^{k}}$ the effective sample size $ess=-ln(\sum_{i=1}^{K}2e^{w_{\rho^{i}}})$ where normalized weights are used for computation and the mean and variance of all weights. Lastly, we save the labels $Y\in R^{2}$ which contain the distance of our current track to the ground truth $\delta_{t}=\sqrt{{\left({\hat{p}}_{t}-{\tilde{p}}_{t}\right)}^{2}}$ and the angular difference between two quaternions $\alpha_{t}=2acos\left({\left({\hat{q}}_{t}*{\tilde{q}}_{t}^{-1}\right)}_{4}\right)$. The two most common problems during tracking are occlusions in the depth image (Figure 4a) by the surgeon and sudden displacements when the body’s position is readjusted for easier cutting (Figure 4b). For our data set, we chose two sequences where either an occlusion or displacement takes place that present considerable problems for the tracking algorithm. Both of the sequences are around 450 frames long. We reran the tracking with the different parameter configurations on both sequences by replaying the aforementioned *rosbag*. An overview of the recorded data set can be seen in Table 2.

### 2.4. Formalization of an Inaccurate Track

We need to define when the tracking is too inaccurate and we consider the track lost in order to train our machine learning models. This is a priori unclear, since suitable choices depend on the current task. The track can be off in its position, which can be measured as the distance to the ground truth, and in its orientation measured as the angle difference to the ground truth. We choose 20 equidistant threshold pairs in order to evaluate different constellations for possible inaccurate tracking, thus enabling a larger flexibility for practical applications
(7)$$\sigma_{i}=\left(\begin{array}{c} d_{i}\\ a_{i} \end{array}\right),\;d_{i}\in \left[0.05,0.2\right]\;a_{i}\in \left[0.13,0.26\right]\;\forall i\in \left[1,2,\ldots ,20\right],$$
where $d_{i}$ represents the tolerance value for the distance (in *m*) to the ground truth and $a_{i}$ the tolerance value for the according angular difference (in rad). The pairs are ordered from very strict ($\sigma_{1}=\left(0.05,\;0.13\right)$) to very tolerant ($\sigma_{20}=\left(0.2,\;0.26\right)$). Tracks that are off by more than $d_{i}$ in distance or $a_{i}$ in angle are considered to be lost in regards to the chosen threshold pair, i.e., one such pair defines the ground truth for the subsequent prediction tasks.

### 2.5. Reject Strategies Based on Machine Learning Methods

We investigate three different approaches for determining when the current track is lost.

*Threshold strategy:* because particle filters are based on probabilistic models, we can implement a simple thresholding based approach, as proposed by Chow [22] as a baseline. Intuitively the maximum log likelihood gives confidence for the current track. Accordingly, we utilize a threshold τ to predict a lost track for a chosen pair of parameters σ=(d,a) characterizing the desired accuracy:(8)$$\hat{ll}<\tau_{\sigma }$$

$\tau_{\sigma }$ could be chosen according to a desired likelihood, yet this requires a suitable scaling of the values and its relation to the geometric accuracy. In our approach, we optimize the threshold value τ that is based on the training data by evaluating a set of candidate values in the range [−200,200].

*Ensemble of binary classification tasks:* we consider two approaches for framing the current problem of deciding whether the current track is lost (as defined by the selected pair $\sigma_{i}$) as a machine learning task. Firstly, we model our problem as a binary classification task for each σ:(9)$$f_{\sigma }:R^{11}⟶\left\{0,1\right\},\;f_{\sigma }\left(x\right)=y$$

Here, the input to the machine learning model *x* is chosen as a vector of representative quantities of the particle filter. It is the four-dimensional vector of the particle filter outputs $X_{\mathrm{out}}$ and the seven parameters $X_{\mathrm{conf}}$, whereby the values are averaged over a 20 frame window, and
(10)$$y=\left\{\begin{array}{cc} 1 & \mathrm{if}\;\exists \delta_{i}\in \left(\delta_{t},\cdots ,\delta_{t+20}\right),\;\delta_{i}>d_{\sigma }\;\mathrm{or}\;\exists \alpha_{i}\in \left(\alpha_{t},\cdots ,\alpha_{t+20}\right),\;\alpha_{i}>a_{\sigma }\\ 0 & \mathrm{otherwise} \end{array}\right.$$

This way, a separate classifier is used for every desired accuracy regarding distance and angle difference. We train our classification models on the same distance and angle difference combinations as in our baseline evaluation and with a 10 frame overlap between windows. We use Support Vector Machine (SVM) [33] with a linear kernel and Random Forests [34] as the classification models. We also apply SMOTE sampling [35] to combat any imbalance in our data set. See [36] for the implementation that we used.

Note that SVM and Random Forests both provide a certainty value of the class by means of the distance to the decision boundary or the percent of trees that predict a certain class, respectively. By shifting the threshold from 0 to a different value, we can extend a model that is optimized for a specific threshold pair σ to neighboring ones. This gives rise to a sparse ensemble of models: instead of using the models $f_{\sigma }$ for all pairs σ, we can rely on a small number of classifiers, which extend a model $f_{\sigma }$ to a neighborhood of different thresholds $\sigma^{\prime }$ by moving the classifier’s threshold ϕ. The range $\sigma^{\prime }$ contains every $\sigma_{i}$ for which a decision threshold ϕ exists, so that classifier $f_{\sigma }^{\phi }$ still yields target values for precision and recall. We then find the minimum number of classifiers, so that their ranges $\sigma^{\prime }$ contain every $\sigma_{i}$ at least once. If a threshold $\sigma_{i}$ is contained in multiple ranges, then the classifier that yields the highest combined precision and recall is chosen. This ensemble of classifiers can then be evaluated on the test set for each threshold combination.

*Regression task:* ss a second modeling approach, we train a regression model:(11)$$f:R^{11}⟶R^{2},\;f\left(x\right)=y$$
where *x* is the same as above and $y=\left(\begin{array}{c} d\\ a \end{array}\right)$ is a prediction of the distance and angle difference. In this case, only one model is trained for every choice of threshold pairs σ. Given a specific threshold pair $\sigma_{i}=\left(d_{i},a_{i}\right)$, a decision is done that is based on the question whether $d_{i}\geq d$ and $a_{i}\geq a$ for f(x)=(d,a) being the predictions of the regression model on *x*. In the following, we evaluate such regression against the same set of distance and angle difference combinations. As candidate models for this regression task, we consider a Linear Regression [37], Support Vector Regression with Gaussian Kernel [38], and Random Forest Regression [39]. All of the models are implemented using the scikit learn library [40].

## 3. Results

All methods are evaluated in a 10-fold cross-validation scheme, adjusted for time series data in order to ensure time consistency: In each split, consecutive data samples amounting to $\frac{1}{11}$ of a time sequence are used for testing while all the preceding samples are used for training. In the next split, the old test samples are added to the training set and the successive $\frac{1}{11}$ of the time series are employed for the next test set, such that every data sample is used for testing at maximum once.

The direct threshold strategy yields a precision larger than 80% but a recall of close to 50% only (see Figure 5). Hence the baseline evaluation shows that the maximum log likelihood is not scaled in such a ways that it allows a robust reject option based on a threshold.

As a comparison, the results obtained by modeling the task as a regression are significantly better. Here, Random Forest regression performs best. It yields both precision and recall close to 90%, which is an acceptable result and much higher than our baseline.

Provided that a separate classification model is designed for every pair of threshold values σ, we can further improve, i.e., using a Random Forest classification, we obtain a precision that is close to 1 and recall of around 98% for almost all values. An overview of all trained models can be seen in Table 3 and a comparison between the best performing regression and classification model against the baseline is shown in Figure 5.

The current classification models, as shown in Table 3, consist of an ensemble of different classifiers according to every pair of threshold values. Because this can become quite exhaustive, the question occurs as to whether an acceptable accuracy can be achieved by using fewer classification models of the same type with different threshold values of the classifier per instance σ. Therefore, we investigate how to minimize the number of different classifiers required in the ensemble given a desired quality of precision and recall, as described in Section 2.5. Table 4 and Table 5 show the results of this method. Interestingly, already a single Random Forest classifier yields precision and recall 0.9, and five models already cover the full space with precision and recalls of 0.98, yielding flexible as well as high performant models for a reject option for particle tracking. For SVM classification, more models are needed for the ensemble. The individual decision boundaries need to be modified, such that target recall values are met, thus resulting in actually lower precision values for a higher number of models.

## 4. Discussion

We have investigated efficient possibilities to enhance particle filtering models for tracking by a reject option, which enables practitioners to react to an alert as soon as the tracking precision is lower than a predefined threshold. Although particle filters provide a tracking probability, this quantity is not scaled in such a way that a simple threshold strategy provides satisfactory results. As a consequence, we have investigated the possibility to model reject options as a learning task, more precisely as regression or classification problem. It turns out that an ensemble of classifiers provides the best results, with an accuracy reaching 0.98, depending on the chosen setting.

The proposed approach provides an immediate possibility to enhance particle filters for tracking in assisted surgery by an alert strategy to prevent malfunctioning. Because of the generality of the proposed approach, we expect that the proposed modeling framework is of more general interest to enhance particle filtering from visual sensor data by a reject option.

## Figures and Tables

**Figure 1 sensors-21-02114-f001:**
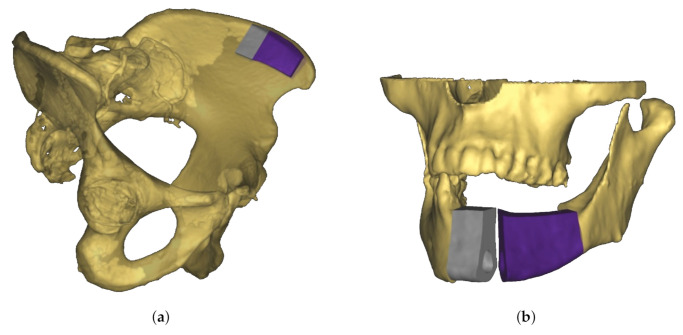
Schematic view of material for jaw reconstruction surgery. (**a**) Three-dimensional (3D) model of graft fitted to patient’s pelvic bone. To reconstruct the geometry of the jaw, two separate transplants are necessary (colored in grey and purple). (**b**) Use of bone graft to reconstruct jaw. Because of the separation into two transplants the bend of the jaw bone can be rebuilt.

**Figure 2 sensors-21-02114-f002:**
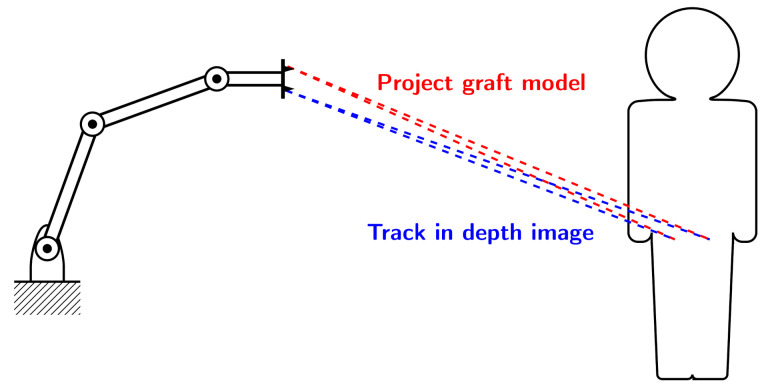
Experimental Setup: 3D camera tracks model of bone graft in depth image of patient’s pelvis (blue). Projector displays cutting lines at tracked position for the surgeon (red).

**Figure 3 sensors-21-02114-f003:**
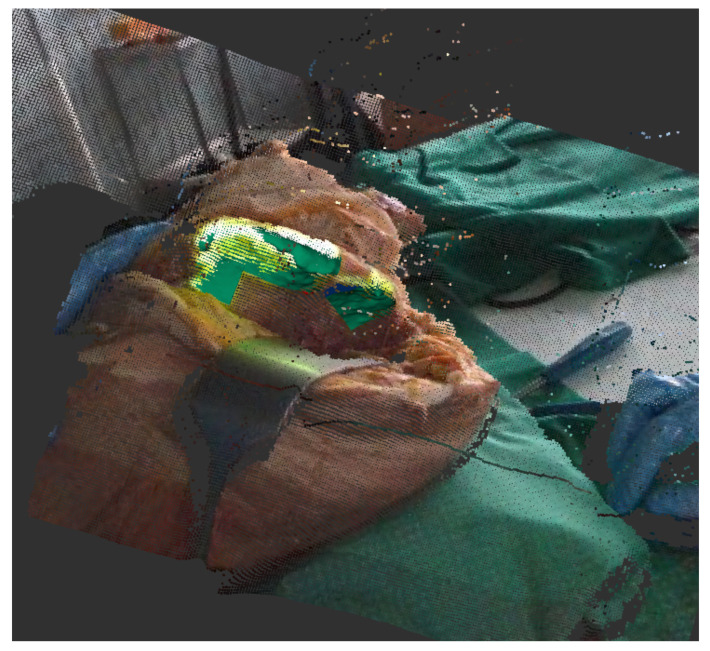
Recorded rosbag of test surgery on corpse shows depth image with a registered color image. The projection on the bone can be seen in light green, the current tracking output in green.

**Figure 4 sensors-21-02114-f004:**
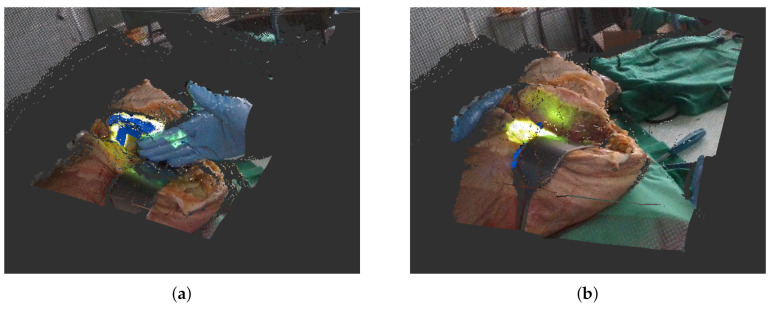
Example settings in which tracking by particle filters faces difficulties. (**a**) Partial occlusion of area to track. Here, a newly recorded track (in blue) is still stable. (**b**) Newly recorded track (in blue) is lost after body was relocated.

**Figure 5 sensors-21-02114-f005:**
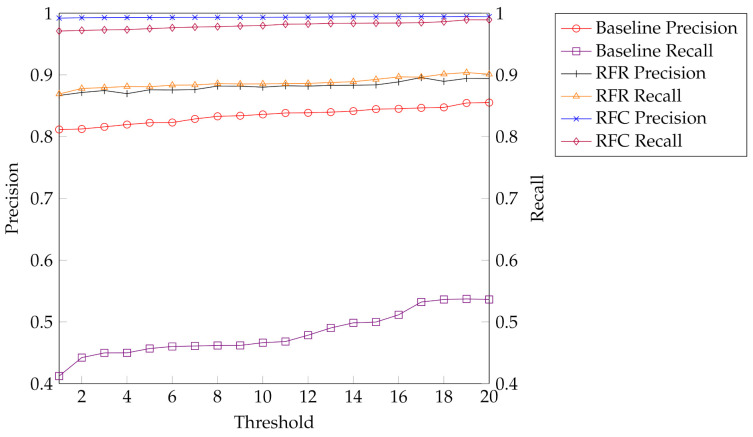
Overview of our training results shows precision and recall for our baseline evaluation, the best performing regression model, Random Forest regression (RFR), and the best performing classification model, Random Forest classification (RFC).

**Table 1 sensors-21-02114-t001:** Parameter configurations. [a,b],step=c would mean the values are chosen between *a* and *b* in steps of *c*.

Translational Noise $n^{\mathrm{trans}}$	Rotational Noise $n^{\mathrm{rot}}$	Velocity Factor vf
[0.0005,0.01],step=0.0005	[0.005,0.1],step=0.005	[0.8,0.99],step=0.01
**Exponential Weight $w^{\exp}$**	**Gaussian Weight $w^{\mathrm{gaus}}$**	**Uniform Weight $w^{\mathrm{uni}}$**
[0.2,0.39],step=0.01	[0.5,0.69],step=0.01	[0.01,0.2],step=0.01
**particle Count K**		
[100,2000],step=100		

**Table 2 sensors-21-02114-t002:** Overview of the recorded data set.

Recorded parameters	7
Recorded outputs	4
Recorded labels	2
No. of different param configurations	8000
Tracking Problems	2
Recording Length	∼450 frames

**Table 3 sensors-21-02114-t003:** This table shows the summarized results for the different instances of machine learning models.

Model	Average Precision	Average Recall
Baseline	0.8344±0.0133	0.4806±0.0357
Random Forest Regression	0.8815±0.0081	0.8877±0.0088
Support Vector Regression	0.8714±0.065	0.9033±0.006
Linear Regression	0.6528±0.0198	0.9612±0.0135
Gaussian Process Regression	0.6612±0.016	0.9315±0.0188
SVM Classification	0.9811±0.013	0.8975±0.0319
Random Forest Classification	0.9935±0.0007	0.9802±0.0055

**Table 4 sensors-21-02114-t004:** This table shows the minimum amount of Random Forest classification models needed to still achieve given goals for precision and recall on the training set and their resulting performance on the test set.

Goal Precision	Goal Recall	No. of Models	Avg. Precision	Avg. Recall
0.98	0.98	5	0.9911	0.9803
0.95	0.95	2	0.9891	0.9742
0.9	0.9	1	0.984	0.9691

**Table 5 sensors-21-02114-t005:** This table shows the minimum amount of support vector classification models that are needed to still achieve given goals for precision and recall on the training set and their resulting performance on the test set.

Goal Precision	Goal Recall	No. of Models	Avg. Precision	Avg. Recall
0.98	0.98	14	0.9701	0.9711
0.95	0.95	7	0.9712	0.9529
0.9	0.9	2	0.9737	0.9012

## Data Availability

The data presented in this study are available on request from the corresponding author. The data are not publicly available due to ethical concerns since they were obtained in a clinical trial.

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
