# Peer review of "Efficient Reject Options for Particle Filter Object Tracking in Medical Applications"

_sensors, 2021, doi:10.3390/s21062114_

Round 1
Reviewer 1 Report
The paper propose the use of machine learning methods for predicting the optimal rejection criteria for object tracking based on particle filter in medical applications. The paper propose to model the task as a regression problem or as an ensemble of classifiers and compare the performance of the proposed method with a likelihood thresholding method. The experimental evaluation is based on an ex-vivo human experiment and it is demonstrating the superior performance of the random forest models.
The paper is clearly written and it is addressing a relevant problem in the medical filed, proposing an interesting solution for automatic detection of inaccurate object tracking. The description of the proposed method is correct and clear.
However, the current manuscript has several weakness:
- The abstract is considering the very wide problem of reliable object tracking even in autonomous manufacturing domain, this field is cited in the abstract but not described in the paper, I suggest the author to focus the revised manuscript on computer assisted surgery.
- The introduction is not able to properly describe the medical motivations of the work and the impact of the proposed solution in the specific clinical filed (jaw reconstruction surgery?). Some medical details are introduced in 2.1, but the overall medical context is not clear.
- The introduction is not able to correctly positioning the proposed method with respect to similar (and recent) works available in literature, limited number of works are cited and limited details are provided (and many works have been published more than 5 years ago). This is also making complex to objectively evaluate the novel contributions of the proposed approaches.
- The considered tracking scenario is not clear (section 2.3), more details about the experimental conditions are required (e.g. tasks performed, overall number of sequences acquired, criteria used for selecting sequences included in the training dataset).
- Moreover, the compliance with ethical standard is not addressed in the paper, and this is an essential point considering that the paper is based on a dataset acquired during a human cadaver experiment.
- The experimental evaluation is based on a single fold validation and no motivation for this choice is provided. More details about the dataset splitting criteria is required, in particular addressing how time consistency is ensured while splitting the dataset.
- I suggest the author to include a more extensive evaluation of the method, for example based on a k-fold cross-validation. It is also necessary to better demonstrate that the proposed methods (in particular RFC) are not overfitting the dataset (as possible description of the performance reported in Fig.5 and table 2).
- The discussions of the results are limited and should be extended to address some interesting findings, for example why a 2 SVMs (Table4) method is outperforming more complex models in terms of avg. precision?
- The quality of Figures could be improved, for example Figure 1 and 2 are not simple to interpret and not correctly described/introduced in the text.
Author Response
We provide our response to the review in the attached pdf file.

Reviewer 2 Report
This paper addresses the question of how to equip a state-of-the art particle filter for tracking by efficient schemes which implement a reliable reject option to detect cases where the tracking is invalid.
The paper is incomplete and is poorly written.
The authors mention that "In a new approach, a robotic arm is equipped with a depth camera to track a representation of the template in the depth image using a particle filter and a projector to display cutting outlines onto the bone (Figure 2) during surgery." Can you present a figure showing a projector that displays cutting outlines onto the bone?
Also, the tracking result of the particle filter is not presented in this paper. Computational load of the tracking based on particle filter needs to be analyzed. Also, the particle filter must run in real time.
Why did you use particle filter? There are other types of filters, such as kalman filters. The computational load of Particle filter may be too large.
The explanation of colored space in Figure 1 is missing.
In (6), why did you use a particle with the maximum log likelihood , instead of the weighted sum?
What is the matrix size in (2)?
The explanations on (3) and (5) are missing.
For instance, the sampling interval dt is omitted in (3).
Why 0.5 is multiplied in (3)?
The explanation on Table 1 is missing. what is "step"?
What do you mean by [0.2, 0.39], [100, 2000],[0.005, 0.1],...?
What is the meaning of "particle count"?
How many particles in the particle filter did you use?
The definition of angular difference alpha_t is not clear.
Author Response

(The authors gave the same response as above.)

Round 2
Reviewer 2 Report
I am satisfied with the revision.